# Design and Analysis of Multi-Layer and Cuboid Coding Metamaterials for Radar Cross-Section Reduction

**DOI:** 10.3390/ma15124282

**Published:** 2022-06-17

**Authors:** Tayaallen Ramachandran, Mohammad Rashed Iqbal Faruque, Mohammad Tariqul Islam, Mayeen Uddin Khandaker, Nissren Tamam, Abdelmoneim Sulieman

**Affiliations:** 1Space Science Centre (ANGKASA), Institute of Climate Change (IPI), Universiti Kebangsaan Malaysia, Bangi 43600, Selangor, Malaysia; tayachandran@gmail.com; 2Department of Electrical, Electronic & Systems Engineering, Universiti Kebangsaan Malaysia, Bangi 43600, Selangor, Malaysia; tariqul@ukm.edu.my; 3Centre for Applied Physics and Radiation Technologies, School of Engineering and Technology, Sunway University, Bandar Sunway, Petaling Jaya 47500, Selangor, Malaysia; mayeenk@sunway.edu.my; 4Department of Physics, College of Sciences, Princess Nourah Bint Abdulrahman University, P.O. Box 84428, Riyadh 11671, Saudi Arabia; nmtamam@pnu.edu.sa; 5Department of Radiology and Medical Imaging, Prince Sattam Bin Abdul Aziz University, Alkharj 16278, Saudi Arabia; a.sulieman@psau.edu.sa

**Keywords:** coding metamaterial, electromagnetic waves, phase response, Radar Cross-Section

## Abstract

This research aimed to develop coding metamaterials to reduce the Radar Cross-Section (RCS) values in C- and Ku-band applications. Metamaterials on the macroscopic scale are commonly defined by effective medium parameters and are categorized as analogue. Therefore, coding metamaterials with various multi-layer and cuboid designs were proposed and investigated. A high-frequency electromagnetic simulator known as computer simulation technology was utilised throughout a simulation process. A one-bit coding metamaterial concept was adopted throughout this research that possesses ‘0’ and ‘1’ elements with 0 and π phase responses. Analytical simulation analyses were performed by utilising well-known Computer Simulation Technology (CST) software. Moreover, a validation was executed via a comparison of the phase-response properties of both elements with the analytical data from the High-Frequency Structure Simulator (HFSS) software. As a result, promising outcomes wherein several one-bit coding designs for multi-layer or coding metamaterials manifested unique results, which almost reached 0 dBm^2^ RCS reduction values. Meanwhile, coding metamaterial designs with larger lattices exhibited optimised results and can be utilised for larger-scale applications. Moreover, the coding metamaterials were validated by performing several framework and optimal characteristic analyses in C- and Ku-band applications. Due to the ability of coding metamaterials to manipulate electromagnetic waves to obtain different functionalities, it has a high potential to be applied to a wide range of applications. Overall, the very interesting coding metamaterials with many different sizes and shapes help to achieve a unique RCS-reduction performance.

## 1. Introduction

The information age that we live in today is also known as the computer age, digital age, or new media age, and it affects the ways individuals communicate, think, and learn. This historical period began in the mid-20th century, and the industrial revolution shifted to information technology. Consequently, technology plays an important role in society and typically has a major impact on daily life. Everyday life becomes easier by adopting new technological inventions, such as wireless communications, satellite communication systems, cloaking devices, and artificial intelligence for industrial applications [1,2,3,4,5,6]. One of the fastest developing fields, known as satellite technology, has become a hot topic among researchers, and its usage is increasing all the time. There have been several bands allocated for satellite applications, but C- and Ku-bands were the focus of this research. Satellites are commonly used in various applications, such as navigation, communication, weather forecasting, radio communication, and broadcasting. ‘C-band’ refers to the portion of the electromagnetic spectrum that exhibits a resonance frequency range from 4 GHz to 8 GHz. Meanwhile, the Ku-band has a frequency range from 12 GHz to 18 GHz, and both bands are primarily used for satellite communications applications.

Many metamaterial-related studies were performed in recent years by constructing various design structures in wide application fields. Faruque et al. proposed a parallel LC-shaped metamaterial resonator for C- and X-band applications [7]. The proposed metamaterial exhibits the optimistic outcome of wider bandwidth-of-transmission coefficient results. Therefore, the proposed design is more compatible and durable for a wide range of microwave applications. Moreover, in 2019, Ahamed et al. proposed a metamaterial with a modified RLC resonator on an FR-4 substrate material [8]. The proposed design can exhibit the larger bandwidth of the negative refractive index (NRI), which is equivalent to 3.40 GHz in the operating region. This phenomenon is possible only when a tunnel structure is added. The magnetic and electrical properties were determined by analysing the effect of tunnelling, which was generated by using the SRR. Additionally, a miniature negative-index metamaterial was constructed, numerically simulated, fabricated, and finally, measured for tri-band applications [9]. The work analyzed the reflection, transmission coefficient, and effective medium ratio values of the proposed metamaterial. This work also adopted CST software by utilising the finite-integration technique to examine the metamaterial’s design. A modified square Z-shaped metamaterial for C- and X-band applications was proposed by Hasan et al. in 2018 [10]. The Z-shaped metamaterial simulated an operating frequency range from 2.0 to 14.0 GHz. The construction of the metamaterial is a square Z-shaped structure with a split that was added to the design and a copper metamaterial structure printed on an FR-4 substrate material. The unit-cell metamaterial has a dimension of 10 × 10 mm^2^, while a 200 × 150 mm^2^ array structure was adopted for the present study.

Besides a wide range of application fields, metamaterials have also been utilised for Radar Cross-Section (RCS) reduction applications. The conventional metamaterial designs have also been utilised in RCS reduction values. RCS is usually employed to evaluate how detectable an object is by radar and is defined as the electromagnetic signature of an object. Moreover, RCS applications are mainly found in military technology, such as aircraft penetrating an enemy area without being detected. If an aircraft is detected, then it will easily be destroyed by any anti-aircraft missiles. Hence, military technology generally uses this RCS application, which has also been used in aviation technology. Therefore, such technology is a very important field of study in radar and antenna engineering. Wen et al. investigated the RCS reduction analysis by proposing a metamaterial absorber (MA) structure that works in the microwave frequency band [11]. The work was validated by measuring conducting-plane and dihedral-angle reflectors coated with both metamaterial and FR-4 layers in an open test site. Moreover, Zhang et al. investigated an RCS reduction in a patch antenna by adopting an MA [12]. The introduced unit-cell MA has a height and thickness of 5.5 mm and 0.3 mm, respectively. A comparison of the patch antenna patterns and reflection coefficients for loading and unloading the metamaterial was presented.

Although metamaterial research has a long tradition, limited coding metamaterial studies for terahertz frequencies have been conducted in recent years. There has been an increased recognition that more attention has been paid to this area of the field since coding metamaterials can manipulate EM waves by adopting various coding sequences with different bits. In 2014, Cui et al. proposed using digital metamaterials in two steps; for instance, they proposed the construction of coding metamaterials by adopting one-bit and introducing unique metamaterial particles that have either ‘0’ or ‘1’ responses controlled by a biased diode [13]. Two-bit coding metamaterials were analyzed in the present work. In 2016, Liu et al. proposed and experimentally demonstrated a transmission-type coding metasurface to bend normally incident terahertz beams in anomalous directions and generate non-diffractive Bessel beams in normal and oblique directions [14]. To overcome the larger reflection and strong Fabry-Pérot resonance that usually originate from a thick silicon substrate, a free-standing design was presented for the coding particle that was formed by stacking three metallic layers with four polyimide spacers alternately. Meanwhile, a similar study in 2017 proposed a tensor-coding metasurface at the terahertz frequency that could take full-state control of an electromagnetic wave in terms of its polarization state, phase and amplitude distributions, and wave-vector mode [15]. Due to the off-diagonal elements that are dominant in the reflection matrix, each coding particle could reflect a normally incident wave to its cross-polarization with controllable phases, resulting in different coding digits. This overall literature review indicates that coding metamaterials have had great impacts on the current technology and require extensive research to explore all the unique properties in many research areas. Since coding metamaterials can manipulate EM waves through just simple coding sequences of unit-cell metamaterial arrangements, one may save time by avoiding and developing many metamaterial structures for an optimization process. The coding metamaterials for C- and Ku-band applications are more suitable to apply and will relatively manifest a better performance than conventional metamaterials.

## 2. Coding Metamaterials for RCS Reduction

Generally, the conventional metamaterials are characterised by continuous values of constituent parameters. However, the coding metamaterials are defined directly by the quantized reflection or refraction phases [16]. The coding digital element has generally changed the way people design, analyse, and understand the metamaterials. This is because it requires only a phase of reflection in a unit-cell design and consequently reduces complexity and the design procedure. Moreover, there is no longer a worry about final coding metamaterial designs that consist of specific unit-cell structures because far-field scattering patterns are influenced by coding sequences. In this chapter, a one-bit coding metamaterials were proposed; outcomes were analyzed based on several parametric studies. During the construction of a unit-cell metamaterial for mimicking the ‘0’ and ‘1’ elements, the main aim was to design an optimised miniaturization structure. Therefore, the goal was to reduce the dimension of a substrate material as long as it produced unique properties. Numerical simulations for coding metamaterials were carried out by adopting well-known CST software. First, the unit cells of coding particles were selected by adopting various parametric design studies. All sets of the various one-bit elements for the optimization process utilised similar substrate materials and dimensions. In this analysis, monostatic Radar Cross-Section (RCS) values from 0 to 18 GHz and a bistatic RCS scattering pattern at 8 GHz (C-band) were analyzed. Second, the phase responses of selected elements were simulated by utilizing the frequency domain solver. Following this analysis, the binary digital elements ‘0’ and ‘1’ were selected, which possess 0 and 180° phase responses. The uniqueness of this type of metamaterial is the distinct responses needed to gain significant phase changes, which provide a great freedom to control EM waves instead of only physically realize digital elements. Once a construction of metamaterial elements was completed, the coding metamaterial structures were analyzed based on several parametric studies. RCS values were calculated based on three different coding sequences based on a number of adopted lattices. Note that the proposed ‘0’ and ‘1’ elements do not need to be described by using macroscopic medium parameters; thus, this work mainly focused on a phase-response property and RCS-reduction value analysis. Both monostatic RCS and 3D bistatic far-field scattering patterns at 8 GHz were investigated for each parametric study. For this analysis, the time domain solver was adopted, and a 0 to 18 GHz frequency range was set.

### 2.1. Development of Unit-Cell Metamaterial Structures

Initially, a unit-cell selection process for the one-bit coding metamaterials was performed through a parametric study of several design structures. All the designs were constructed with similar dimensions of 5 × 5 mm^2^ substrate materials. The trial-and-error method was utilised to discover the optimised unit-cell metamaterial design for the coding metamaterial application. The focus on the designs was divided into four structures, namely circular, square, triangular, and unique shapes of SRR metamaterials. In each subsection, various substrate materials were used to analyse the outcomes, such as FR-4, Rogers RT6002, and Rogers RO3006, which were bought from Shenzhen Dihe Electronic Co., Ltd. (Shenzhen, China). The selected substrate materials have dielectric constants and tangent losses of 4.3, 2.94, and 6.5 and 0.025, 0.0012, and 0.002, respectively. These substrate materials were adopted because they possess large differences in dielectric constant values between the materials. Therefore, they will typically reveal whether the dielectric constant of each substrate material influences the RCS-reduction values. Moreover, these substrate materials are easily available in the local market.

Three designs, as illustrated in Table 1, were adopted, and the bistatic scattering patterns at 8 GHz (C-band) of each design were tabulated as well in the same table. This analysis revealed that most of the design structures exhibit similar bistatic RCS peak points, which range from −55.1 dBm^2^ to −58.9 dBm^2^. The overall designs exhibited the lowest bistatic RCS values when an FR-4 substrate material was used compared to other materials. Moreover, the scattering patterns exhibited equivalent outcomes for all the design structures in spherical shapes. Figure 1 demonstrates the monostatic RCS values of each design with various substrate materials. There were only minor discrepancies that occurred between the designs and the substrate materials.

#### 2.1.1. Circular Split-Ring Resonator

The monostatic RCS values happened between a common range from −110 dBm^2^ to −40 dBm^2^. Moreover, only minor differences were visible between all the substrate materials in each design structure. It was revealed that better RCS-reduction values are possible when utilizing the second-highest dielectric constant substrate material. This indicates that the dielectric properties of material structure do not influence the outcomes and the unique metamaterial properties are possible by adopting a distinct design structure. Meanwhile, the Rogers RT6002 substrate material produces the second better RCS values, although it has a 2.94 dielectric constant. In this circular metamaterial structure, the thickness of substrate material influences the outcomes, whereas the RCS value increases as the thickness of the substrate material decreases. The thicknesses of the substrate materials adopted in this work were 1.6 mm, 1.524 mm, and 0.25 mm, respectively.

#### 2.1.2. Square Split-Ring Resonator

Three types of simple square metamaterial designs were constructed, and performance analyses were carried out for each design structure. Meanwhile, three similar types of substrate materials, namely FR-4, Rogers RT6002, and Rogers RO3006, were adopted in this work. Based on the circular unit-cell metamaterial design, the designs each exhibited a similar performance in terms of scattering patterns and monostatic RCS values. However, a square unit-cell design exhibited bistatic RCS values at 8 GHz with the maximum and minimum values ranging from −53 dBm^2^ to 58.9 dBm^2^, as demonstrated in Table 2. The FR-4 substrate material exhibited the lowest value of bistatic RCS as well. Moreover, the square metamaterial design enabled slightly lower monostatic RCS values that reached less than −40 dBm^2^, as illustrated in Figure 2. As discussed earlier, the thickness of the substrate material also influenced the RCS values for this type of unit cell.

#### 2.1.3. Triangular Split-Ring Resonator

Similar substrate materials and dimensions were also utilized for this triangular design. A comparison revealed unusual bistatic RCS peak values at 8 GHz when the thickness of the substrate material decreased. Design 2 manifested a peak point that increased at first when the substrate material changed from FR-4 to Rogers RT6002. However, the value slightly decreased to −60.3 dBm^2^ when the Rogers RO3006 substrate material was used (as demonstrated in Table 3). It showed that besides substrate material thickness, design structure also influenced the outcomes. Moreover, this design structure possesses monostatic RCS values similar to the circular design that range from −110 dBm^2^ to −40 dBm^2^, as shown in Figure 3.

#### 2.1.4. Unique Shape

In this section, unique combinations of square and circular designs are discussed that were created with similar dimensions. Instead of only circular or square metamaterial design structures, these unique designs exhibit unusual scattering properties and RCS values, as illustrated in Table 4 and Figure 4. This type of design structure enabled the lowest bistatic RCS peak point at 8 GHz of −51.7 dBm^2^ for Design 1 with the FR-4 substrate material. Moreover, the result was the lowest compared to the rest of the designs. Unusual scattering-pattern peak points were gained when the substrate thickness was reduced. For example, Design 1 and Design 2 exhibited similar flows of changes wherein the peak point increased and was then reduced. Meanwhile, the result of Design 3 decreased at first and then increased back. All the designs for the three substrate materials manifested unique monostatic RCS resonance patterns that ranged from 10 GHz to 18 GHz, respectively. The combination of square and circular shapes enables these unique patterns to occur, and the other single shapes manifest only incline graphs at the stated frequency range. Therefore, from this analysis, one can conclude that a combination of several shapes in a unit-cell metamaterial design enables better outcomes and will be suitable for the proposed objectives. Moreover, the adopted substrate materials with various dielectric constant values and thicknesses in each design structure manifest almost similar bistatic scattering patterns. Hence, this analysis revealed that the reduction ability is not only based on these parameters but also requires a combination of adopted substrate materials and a proposed design (a metamaterial design in this context) to exhibit unique properties. However, further reduction values are possible when the selected unit cells are arranged in the coding sequences. Furthermore, after the unit-cell selection process, the RCS-reduction ability in coding metamaterial is influenced by how the coding sequences are constructed based on the number of lattices.

### 2.2. Elements ‘0’ and ‘1’ of One-Bit Coding Metamaterials

For the optimization analyses in the above subsection, two unit cells were adopted to be assigned as ‘0’ and ‘1’ elements. The FR-4 substrate material without any metamaterial structure was adopted as element ‘0’, while Design 1, with a combination of square and circular metamaterial designs, was adopted as element ‘1’. Figure 5a,b demonstrates the two unit cells selected for the coding metamaterial design. All the descriptions of the unit cells are tabulated in Table 5. Since the unit cells in one-bit coding metamaterials possess 0 and π phase responses, the selected unit cells were based on this criterion. Meanwhile, Figure 5c illustrates the phase responses and phase differences of elements ‘0’ and ‘1’. The elements have a phase response difference of 180° at a frequency range from 7.07 GHz to 8.00 GHz (C-band) and 14 GHz to 18 GHz (Ku-band). However, the absolute phase response of the proposed element ‘0’ may not exhibit a zero response at a specific frequency. It will not affect any physics because the phase can be normalized to zero. Although the phase-response properties have been seen in specific ranges, which were from 7.07 GHz to 8.00 GHz and 14 GHz to 18 GHz, this work used a frequency range from 0 to 18 GHz to analyse the alterations in the responses. The phase response properties were also validated by utilizing High-Frequency Structure Simulator (HFSS) software. The phase-response properties from the HFSS software revealed almost similar results compared to the CST software, as illustrated in Figure 5c. Therefore, the monostatic RCS calculation for the proposed coding metamaterials is suitable to be applied to C- and Ku-band applications. However, the reflection coefficient results of both elements are demonstrated in Figure 5d. Element ‘1’ exhibited peak points at approximately two frequency ranges: from 5.7 GHz to 8.0 GHz and 11.9 GHz to 15.9 GHz. Moreover, the first resonance curve fell only in the C-band, while the second curve manifested in two frequency bands—likely X- and Ku-bands. The element ‘0’ exhibited an almost straight line near 0 dB for the whole frequency range. Note that the phase response of element ‘1’ had a 180° phase shift when the first reflection-coefficient resonance moved to a second resonance.

### 2.3. Coding Metamaterial Parametric Study

Coding metamaterials can manipulate EM waves, whereby a normally incident beam can reflect in more than one oriented direction under various coding sequences. To assess the aforementioned physical phenomenon, a general square metamaterial design composed of N × N equal-sized lattices was adopted by occupying the proposed elements. Then, the adopted 1-bit coding metamaterials were investigated and analyzed based on five different numbers of lattices: 4, 6, 8, 12, and 20, respectively. Each analysis of the lattices comprised three different coding sequences, and the obtained RCS values were compared. Therefore, an arrangement of the elements ‘0’ and ‘1’ based on the number of lattices was designed by adopting the unit cells selected in the previous subsection. Moreover, every coding sequence was designed with distinct patterns to analyse the performance of RCS reduction. For each number of lattices, the arrangement of coding sequences in each row was tabulated. Meanwhile, the coding metamaterial design for 20 lattices was illustrated in only graphical form. The bistatic scattering patterns of the proposed coding metamaterials at 8 GHz were also analyzed in this work.

#### 2.3.1. Four Lattices

First, the four lattices were adopted, and outcomes were analyzed based on the monostatic RCS values and scattering patterns of three different coding sequences, as shown in Figure 6 and Table 6. The analysis revealed that the scattering patterns do not have major discrepancies when compared to the unit-cell metamaterial structure, as in the above subsection. However, the maximum bistatic RCS values reduced tremendously. The element ‘1’ had a peak point of −51.7 dBm^2^, while the proposed coding metamaterials were reduced to −27.8 dBm^2^ with a similar spherical scattering pattern. From this analysis, one can clearly see that the distinct coding sequence can manipulate the RCS values and scattering pattern. Moreover, considering the three different sequences, Coding Sequence 1 exhibited a slightly better performance.

#### 2.3.2. Six Lattices

Figure 7 graphically illustrates the monostatic RCS values and bistatic scattering patterns of the coding metamaterials with six lattices. Meanwhile, the coding sequences, row by row, are tabulated in Table 7. The scattering patterns started to show some changes over the shapes in all three coding sequences. Coding Sequence 2 had the lowest bistatic RCS peak points at −19.5 dBm^2^. Meanwhile, Coding Sequences 1 and 3 exhibited maximum reduction values of −20.6 dBm^2^ and −21.3 dBm^2^, respectively. A comparison of four lattices and six lattices revealed promising outcomes; for example, at C-band, the proposed coding metamaterials exhibited monostatic RCS reduction values between approximately −43 dBm^2^ and −30 dBm^2^ and between approximately −32 dBm^2^ and −22 dBm^2^, respectively. This revealed that the increasing number of lattices had a great impact on the monostatic RCS values. Therefore, larger coding metamaterial designs will have better RCS reduction properties compared to smaller designs.

#### 2.3.3. Eight Lattices

Only slight differences were visible for monostatic RCS values when compared to the six-lattice coding metamaterial designs, as demonstrated in Figure 8. Moreover, the three coding sequences exhibited almost identical RCS values for the whole 0 to 18 GHz frequency range. However, the scattering patterns of these sequences were manipulated and exhibited the lowest bistatic RCS value of −20.7 dBm^2^ for Coding Sequence 3. It was proven that the scattering patterns can be controlled by adopting various coding sequences. This will save time by writing coding sequences; otherwise, researchers must develop a new design if the constructed metamaterials fail to manifest desired properties. Table 8 illustrates the eight-lattice coding sequences row by row.

#### 2.3.4. Twelve Lattices

The 12 lattices exhibited the second-best and unique results in this parametric study. Overall, all the coding sequences successfully produced monostatic RCS values that reached almost 0 dBm^2^ at the Ku-band, as shown in Figure 9. Meanwhile, at the C-band, RCS values ranging from −20 dBm^2^ to −9 dBm^2^ occurred for all the metamaterial sequence designs. However, the bistatic scattering patterns of the three coding sequences differed from each other. Moreover, the peak value of bistatic RCS reached a −8.3 dBm^2^ for Coding Sequence 3. A comparison of the coding sequences in the 6 and 12 lattices revealed tremendous changes in the monostatic RCS values. Therefore, for one-bit coding metamaterials, a larger design structure is required to gain the highest RCS reduction values. The arrangement of elements ‘0’ and ‘1’ for the 12 lattices is illustrated in Table 9.

#### 2.3.5. Twenty Lattices

Lastly, the 20-lattice coding metamaterial designs were analyzed; the adopted coding sequences are illustrated in Figure 10a–c. The monostatic RCS-reduction values exhibited similar patterns for all the coding sequences, as shown in Figure 11. At the C-band, the exhibited RCS values occurred in a range from −10.4 dBm^2^ to −0.7 dBm^2^, while the Ku-band was able to reach positive values in a range from 0.05 dBm^2^ to 6.3 dBm^2^, respectively. Although all the proposed coding sequences exhibited almost equivalent curves, the bistatic scattering patterns of the designs manifested unique results. The bistatic RCS had a peak point of −0.142 dBm^2^ for Coding Sequence 2, and the result varied based on the proposed sequences compared to the other two coding metamaterials.

Overall, five lattices revealed promising outcomes, and the monostatic RCS values were tremendously reduced for the larger coding metamaterial design. As mentioned earlier, the one-bit coding metamaterial design utilised only the ‘0’ and ‘1’ elements and arrange distinct in dimensions based on the number of lattices. Therefore, initially, a smaller number of lattices (four) was adopted to analyse the performance changes in RCS values. Moreover, the first three numbers of lattices increased by two, while the following numbers of lattices increased by four and eight, respectively. The changes in the monostatic RCS values from 6 to 12 lattices manifested the first tremendous outcomes and thus revealed better results compared to the previous numbers of lattices. However, the 20-lattice coding metamaterial designs exhibited extraordinary RCS reduction values, even enabling positive values. The monostatic RCS values almost reached 6.5 dBm^2^ with the 20-lattice coding metamaterial designs. Thus, the coding metamaterial enabled unique properties by simply arranging the selected elements based on the desired dimension and controlling scattering patterns, but for optimized RCS-reduction values, a larger one-bit coding metamaterial is required.

### 2.4. Effect of Multi-Layer Structure for RCS Reduction

Since the existing literature on the coding metamaterial field focuses only on the conventional lattice forms of coding design structures, this study introduced unique structures, such as multi-layer designs and cuboid designs, by adopting one-bit coding elements. Therefore, the focus of this parametric study was to identify the changes in RCS values when adding multiple layers of coding-based metamaterial structures. A 1-bit 12-lattice design was adopted for this analysis study. The best among the three different coding sequences was selected for this case study. Double and triple layers of the proposed coding sequences were utilized to calculate the RCS values. The one-bit coding metamaterials for both layers exhibited better monostatic and as well bistatic RCS values compared to the single-layer structure, as shown in Figure 12a,b. The bistatic RCS values reached peak points, such as −7.87 dBm^2^, −9.02 dBm^2^, and −7.92 dBm^2^, respectively, for all three coding sequences. Meanwhile, the triple-layered structure also exhibited lower RCS values, which were reduced by 25% from the double-layered structure.

### 2.5. Cuboid Coding Metamaterial Design

Cuboid coding metamaterial designs were proposed in this research; an analysis was conducted by performing comparisons between the numbers of lattices from 4 to 12. For each lattice, the best-performing coding metamaterial designs among three were selected for this analysis. Examples of the cuboid design structures for the one-bit coding metamaterials are illustrated in Figure 13a–d. The monostatic RCS values of the adopted lattices are demonstrated in Figure 14; they indicate uniform RCS-reduction values when the size of the cuboid increases based on the number of lattices. Moreover, uniform reductions in the bistatic RCS values are demonstrated in Figure 15a–d. The flow of the values was as follows: −17.1 dBm^2^, −12.7 dBm^2^, −7.6 dBm^2^, and −4.36 dBm^2^, respectively. Therefore, it can be clearly seen that RCS value was inversely proportional to the size of the proposed coding metamaterials. In a nutshell, a comparison of a cuboid design with a lattice form indicated that almost 65% of reduction value was successfully at 8 lattice coding design. 

### 2.6. Comparison between Conventional and Unique Coding Metamaterial Structures

As indicated in this section, the construction of coding metamaterials relies on the most important characteristics, known as digital characterizations of every unit-cell design by using binary codes. This study adopted one-bit coding metamaterials, and two types of unit cells were introduced to mimic the ‘0’ and ‘1’ elements, which possess opposite phase reflections of 0° and 180°. The same elements were incorporated in a unique-structure coding metamaterial analysis. The arrangement of these coding particles in desired sequences on a two-dimensional plane could manifest certain functionalities. In this study, analytical investigations were performed to accurately estimate the radiation pattern of the proposed coding particles in N × N equal-sized lattices. Besides conventional chessboard-like configurations, several unique configurations for coding metamaterial designs were also adopted in this work. The construction of these configurations with a 0 phase created a perfect magnetic conductor, while a 180° phase possessed a combination of perfect magnetic and electrical conductors. Hence, the reflections of any normally incident EM wave will typically cancel out, leading to a reduction in RCS values. For instance, the conventional coding metamaterial designs with six lattices were the first to produce the different basic functionalities of bistatic RCS scattering patterns with interesting coding sequences. Moreover, all the coding sequences with six lattices generated a normally incident plane wave that was diverted from a spherical shape to a several-directions scattering pattern, as illustrated in Figure 7. Furthermore, the coding patterns in the following number of lattices exhibited several scattering-pattern directions instead of spherical responses in the lower lattices. Therefore, the best reduction values can be gained by optimising the coding sequences in lattice forms.

In general, the radiation pattern was always symmetrical in terms of the x-z or y-z axes for the one-bit coding metamaterial designs [17]. Hence, it was difficult to realize asymmetrical radiation patterns using one-bit coding metamaterials in a lattice form. Researchers opt for two-bit or higher bit design structures to gain unique scattering beam directions. Nevertheless, the proposed multi-layer cuboid metamaterial designs in this study were able to manifest asymmetrical radiation patterns, indicating an anomalous direction caused by the gradient phase distribution. This is because these unique structures have gradient phase changes and are based on Snell’s law; normally incident waves are reflected at oblique angles, which was confirmed by using the simulation analysis of far-field scattering patterns, as shown in Figure 12 and Figure 15. Hence, this unique phenomenon was a novelty of this research. Besides that, Coding Sequence 3 with 8 lattices manifested a slightly higher reduction value compared to the six-lattice design, as shown in Table 10. This clearly indicates that an increase in the number of lattices does not influence the RCS reduction value. A unique combination of coding sequences with specific lattices will greatly encourage the RCS value to reduce. However, the triple multi-layer coding metamaterial analysis on 12 lattices indicated a 23% higher reduction value when compared to conventional designs, and the value was further reduced by adopting cuboid coding metamaterials. As shown in Table 10, using from 4 to 12 lattices, the conventional and cuboid coding metamaterials enabled reduction values of approximately 70.1% and 74.5%, respectively. Hence, the results prove that the cuboid coding metamaterial designs exhibited better reduction values compared to the conventional or multi-layer coding metamaterial designs.

## 3. Conclusions

In this work, the authors presented an integration of fast-growing coding metamaterials for RCS-reduction applications at the C- and Ku-bands. The elements for the coding metamaterials were selected through the trial-and-error method by constructing various design structures, such as circular and square metamaterials. The selected elements, namely ‘0’ and ‘1’, for one-bit coding sequences were based on phase responses. Although all the numerical simulations were conducted using CST software, the phase-response properties were validated through HFSS software. The single-layer coding metamaterial structure revealed corresponding results wherein monostatic RCS values reduced as numbers of lattices increased. Multi-layer cuboid coding metamaterials were also introduced in this work. The one-bit coding metamaterials with multi-layer structures successfully exhibited further RCS reduction values when compared to single-layer structures. The cuboid coding metamaterials demonstrated a better reduction value and is very appropriate to be applied to the miniaturization concept that requires further RCS-reduction properties. These proposed unique structures can be utilized in smaller devices since they can further reduce the RCS value instead of using larger-sized coding metamaterials. Coding metamaterials are generally new to the scientific field, and it can be applied to a wide range of applications, such as manipulating antennas’ radiation beams, reducing the scattering features of targets, and manifesting the various extraordinary properties of metamaterials. Moreover, RCS reductions generally have a crucial role in stealth technology for aircraft, missiles, ships, and other military vehicles. With smaller RCSs, vehicles can evade radar detection more effectively, whether it be from land-based installations, guided weapons, or other vehicles. Overall, the results obtained from these analyses give a clear view of coding metamaterials’ integration effects for proposed frequency bands. Coding metamaterials have better RCS-reduction values; hence, it is suitable for C- and Ku-band applications. A further investigation into coding metamaterial will be a valuable asset in this technological era since the materials possesses unique monostatic RCS and bistatic scattering-pattern behaviours.

## Figures and Tables

**Figure 1 materials-15-04282-f001:**
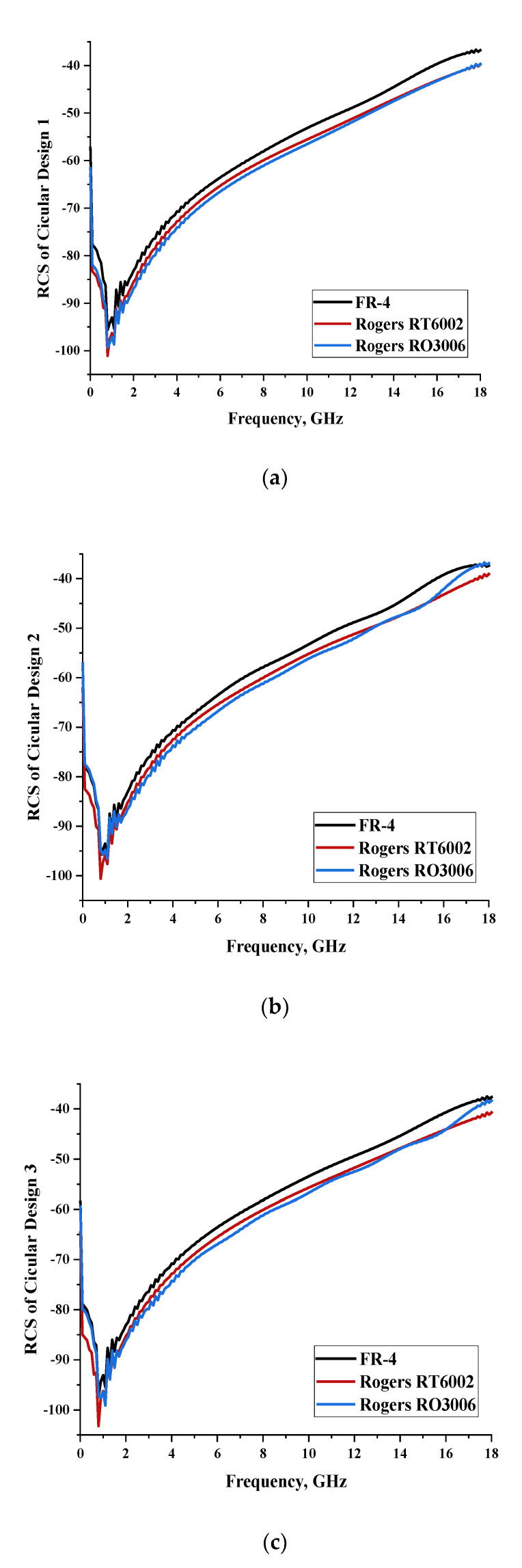
Monostatic RCS values for circular metamaterial unit-cell design with various substrate materials: (**a**) Design 1; (**b**) Design 2; and (**c**) Design 3.

**Figure 2 materials-15-04282-f002:**
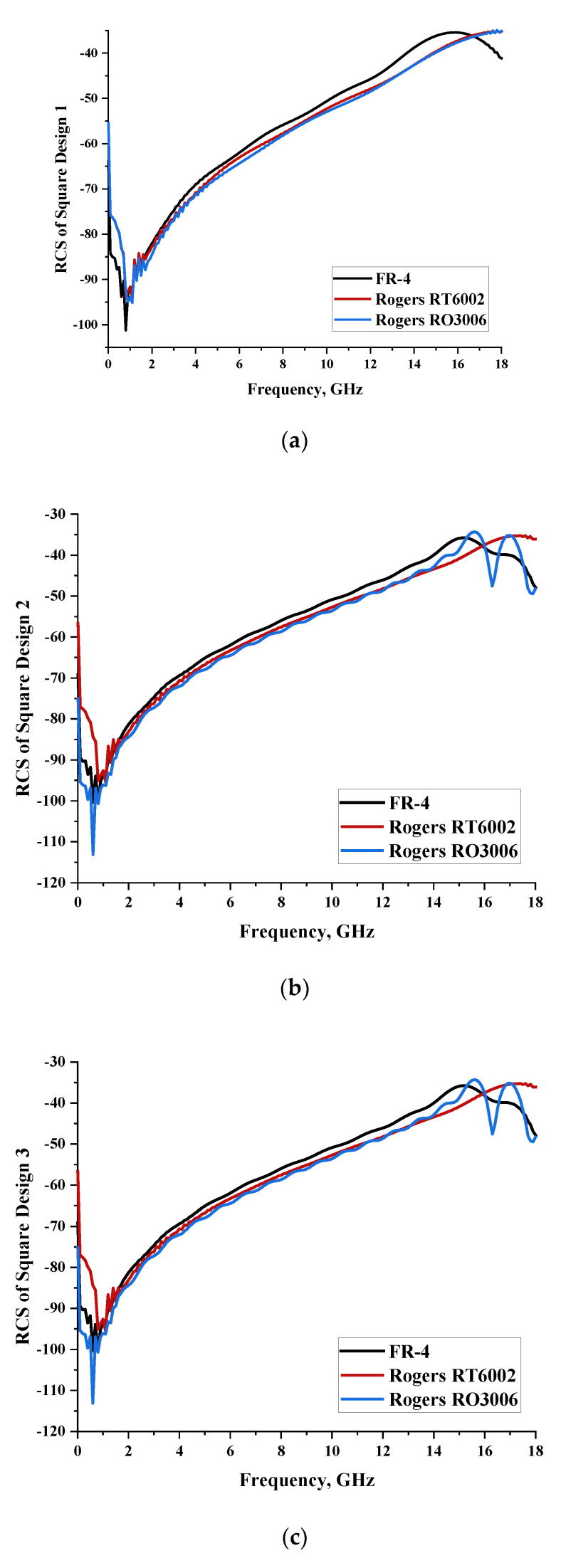
Monostatic RCS values for square metamaterial unit-cell design with various substrate materials: (**a**) Design 1; (**b**) Design 2; and (**c**) Design 3.

**Figure 3 materials-15-04282-f003:**
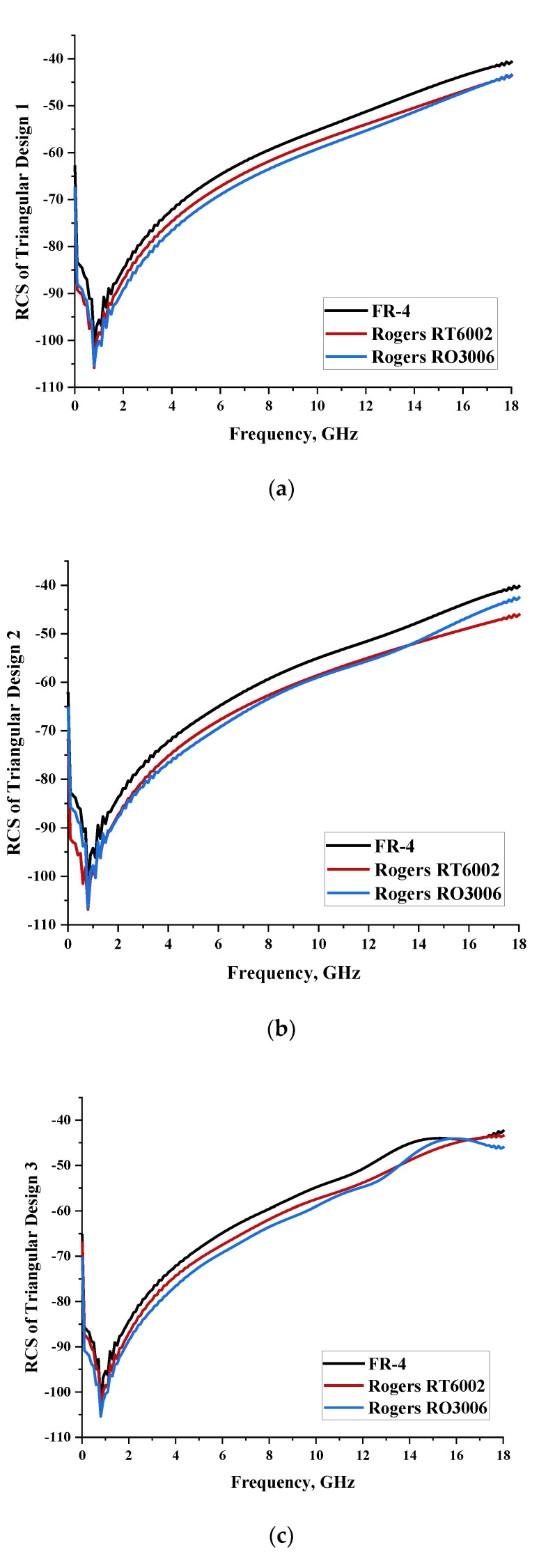
Monostatic RCS values for triangular metamaterial unit-cell design with various substrate materials: (**a**) Design 1; (**b**) Design 2; and (**c**) Design 3.

**Figure 4 materials-15-04282-f004:**
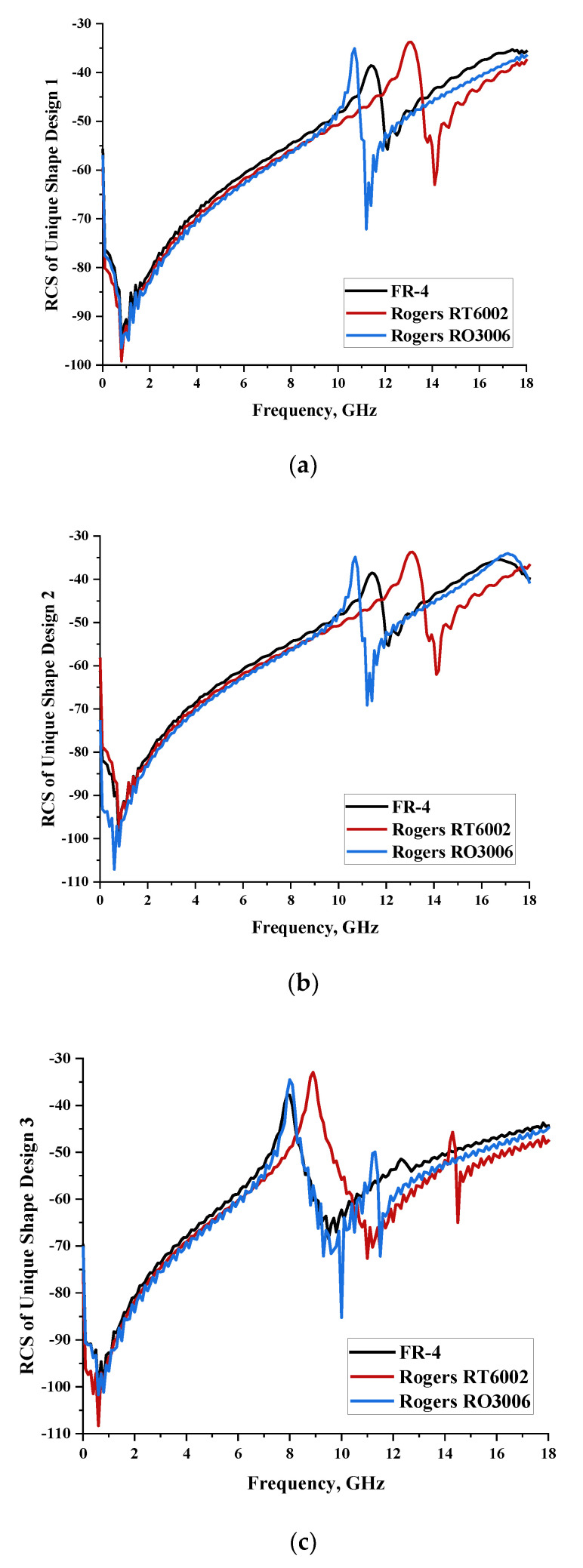
Monostatic RCS values for uniquely shaped metamaterial unit-cell design with various substrate materials: (**a**) Design 1; (**b**) Design 2; and (**c**) Design 3.

**Figure 5 materials-15-04282-f005:**
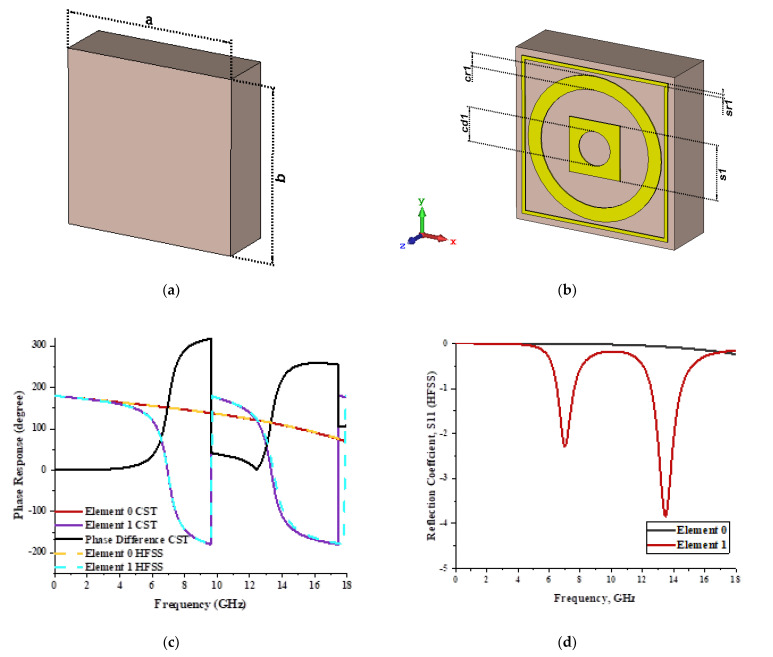
One-bit coding metamaterials with: (**a**) element ‘0’; (**b**) element ‘1’; (**c**) phase responses; and (**d**) S11 of both elements.

**Figure 6 materials-15-04282-f006:**
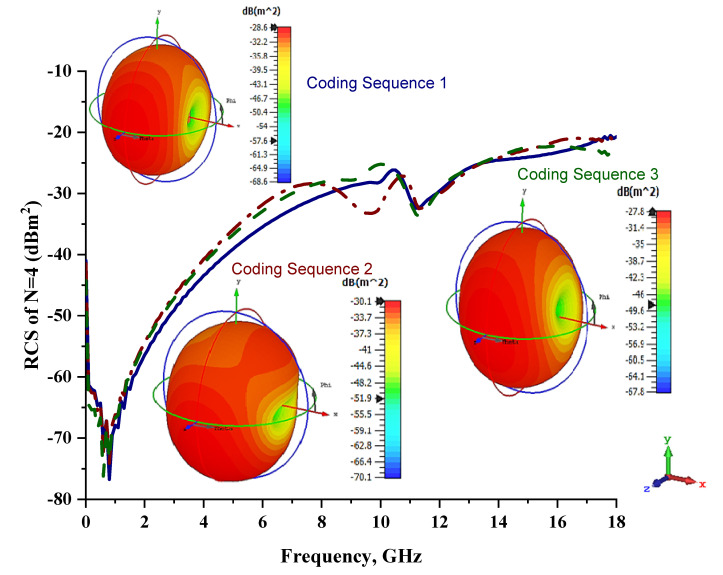
Monostatic RCS and bistatic scattering patterns (at 8 GHz) of three different coding metamaterials with four lattices.

**Figure 7 materials-15-04282-f007:**
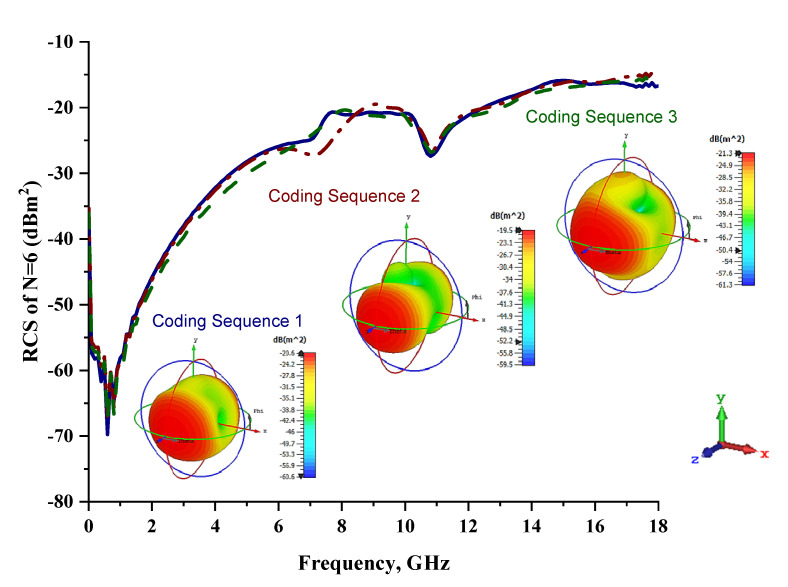
Monostatic RCS and bistatic scattering patterns (at 8 GHz) of three different coding metamaterials with six lattices.

**Figure 8 materials-15-04282-f008:**
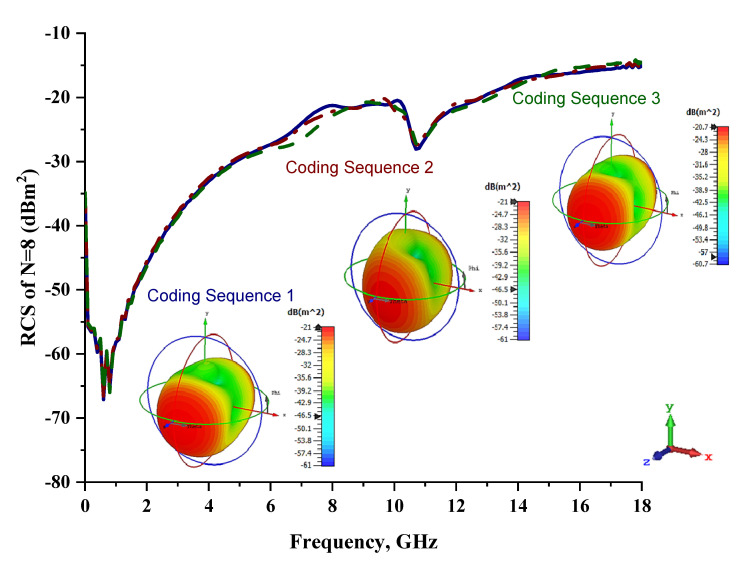
Monostatic RCS and bistatic scattering patterns (at 8 GHz) of three different coding metamaterials with eight lattices.

**Figure 9 materials-15-04282-f009:**
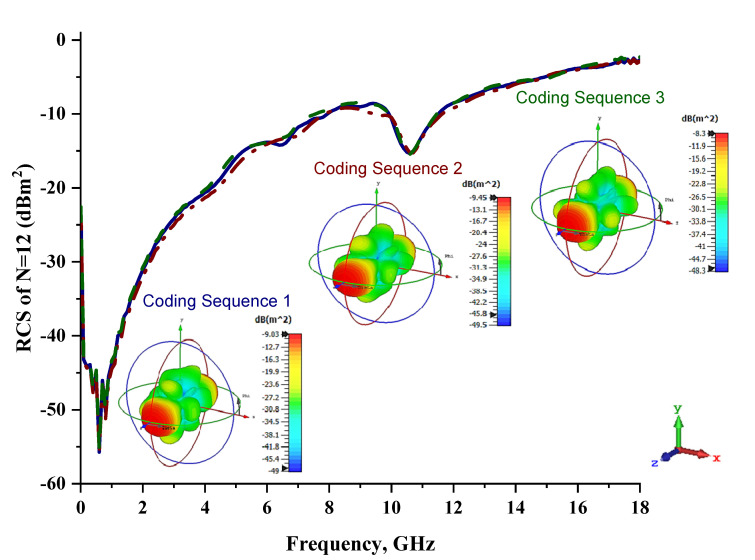
Monostatic RCS and bistatic scattering patterns (at 8 GHz) of three different coding metamaterials with 12 lattices.

**Figure 10 materials-15-04282-f010:**
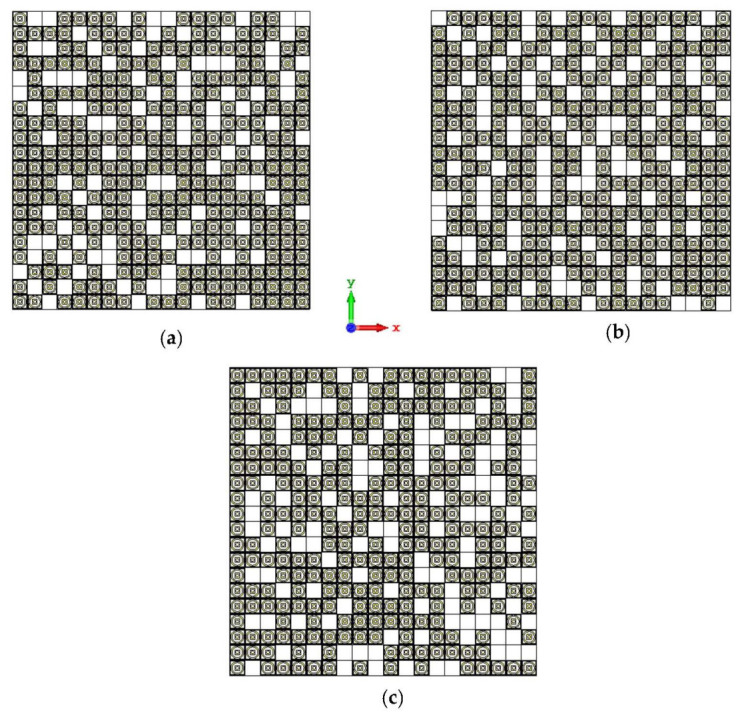
Coding metamaterial designs for 20 lattices: (**a**) Coding Sequence 1; (**b**) Coding Sequence 2; and (**c**) Coding Sequence 3.

**Figure 11 materials-15-04282-f011:**
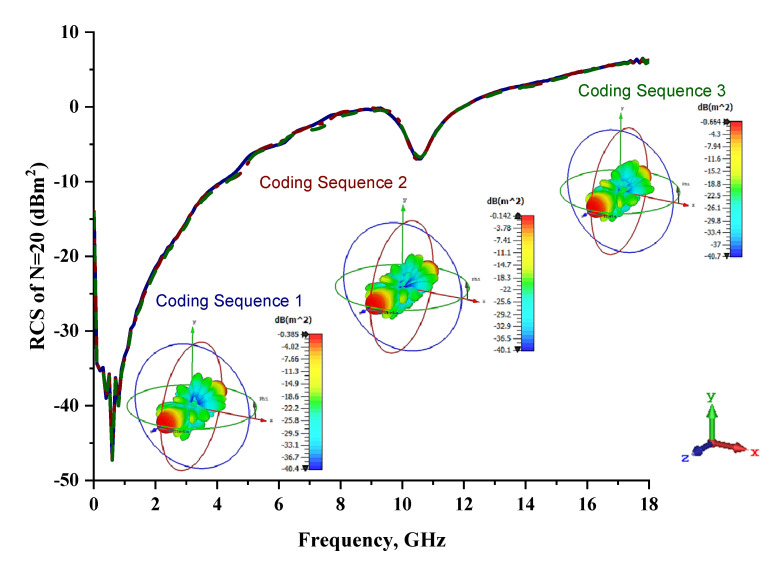
Monostatic RCS and bistatic scattering patterns at 8 GHz for three different coding metamaterials with 20 lattices.

**Figure 12 materials-15-04282-f012:**
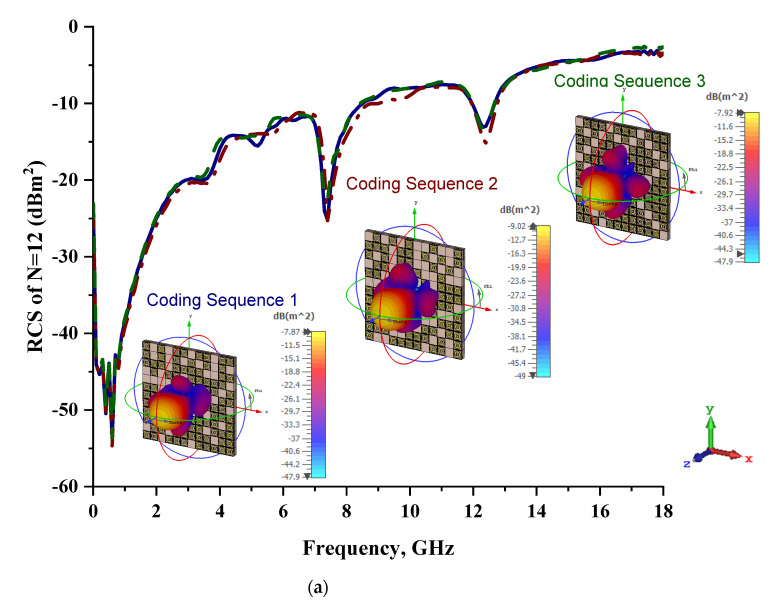
Monostatic RCS values and bistatic RCS values at 8 GHz for one-bit coding metamaterials with (**a**) double layers and (**b**) triple layers.

**Figure 13 materials-15-04282-f013:**
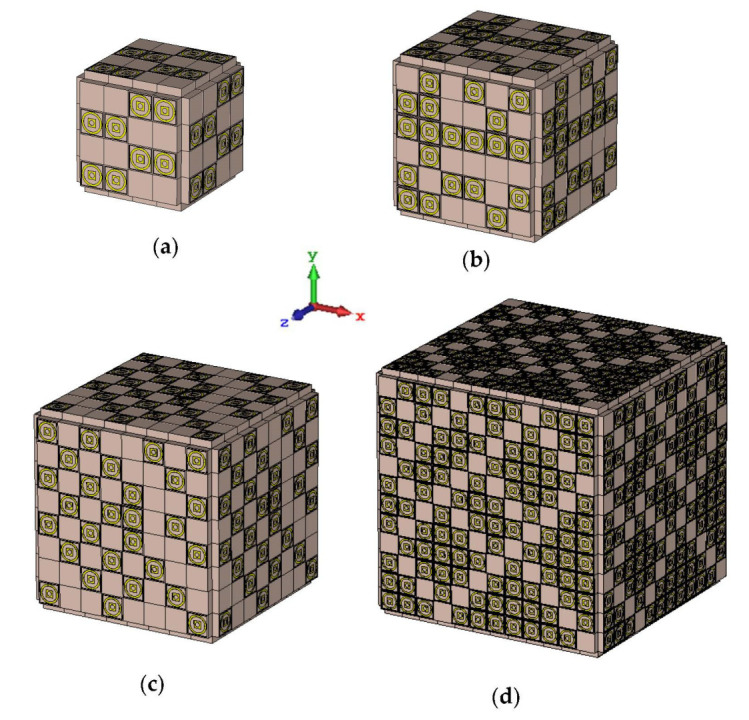
Cuboid coding metamaterial structures of (**a**) Coding Sequence 3 with 4 lattices; (**b**) Coding Sequence 2 with 6 lattices; (**c**) Coding Sequence 3 with 8 lattices; and (**d**) Coding Sequence 3 with 12 lattices.

**Figure 14 materials-15-04282-f014:**
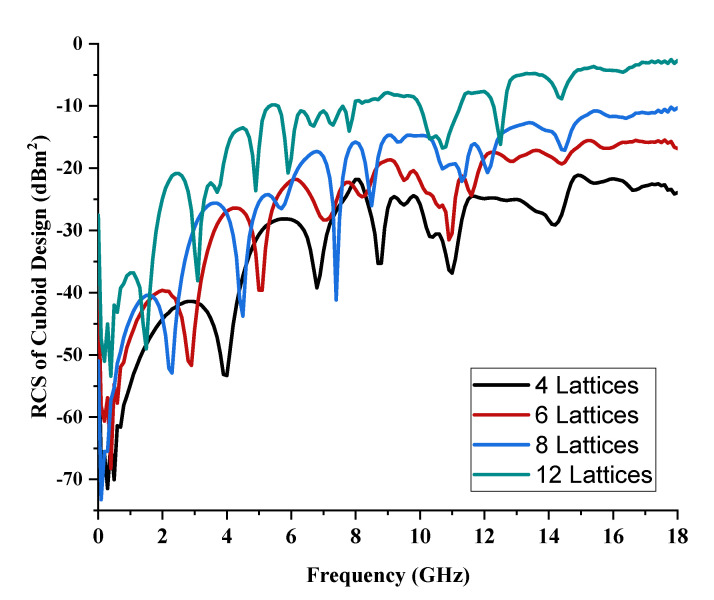
Monostatic RCS data of one-bit cuboid design.

**Figure 15 materials-15-04282-f015:**
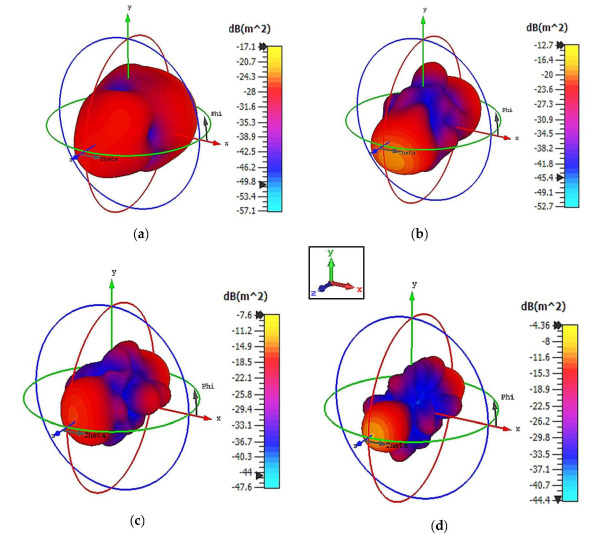
Bistatic scattering patterns at 8 GHz for (**a**) 4 lattices; (**b**) 6 lattices; (**c**) 8 lattices; and (**d**) 12 lattices.

**Table 1 materials-15-04282-t001:** Bistatic scattering patterns of circular structure with the various substrate materials.

Designs	FR-4	Rogers RT6002	Rogers RO3006
Design 1 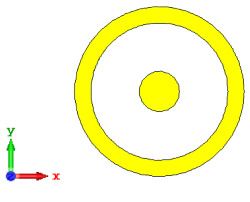	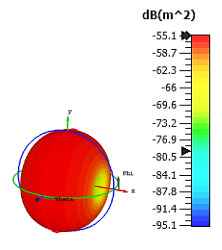	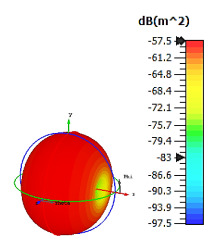	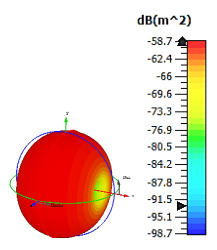
Design 2 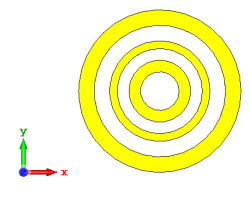	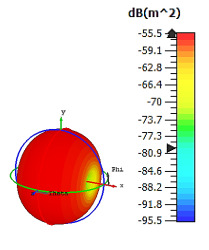	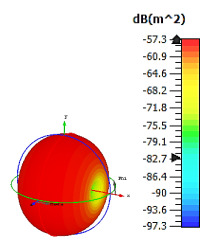	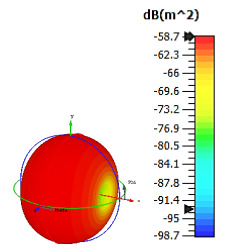
Design 3 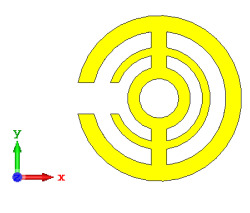	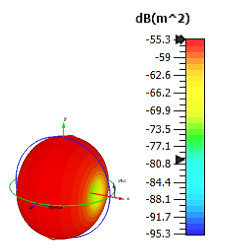	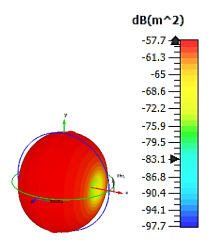	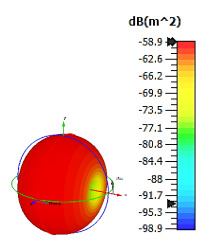

**Table 2 materials-15-04282-t002:** Bistatic scattering patterns of square structure with the various substrate materials.

Designs	FR-4	Rogers RT6002	Rogers RO3006
Design 1 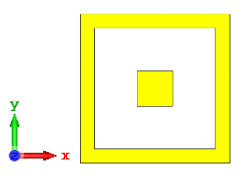	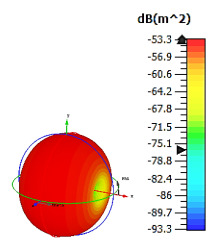	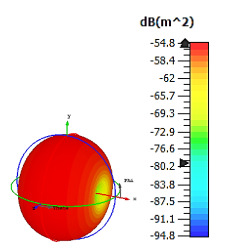	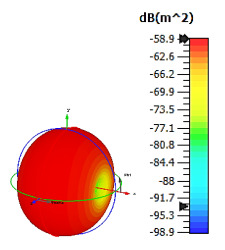
Design 2 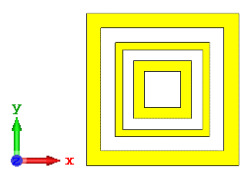	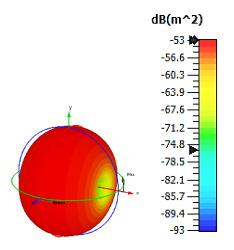	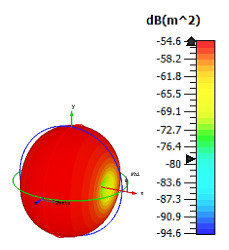	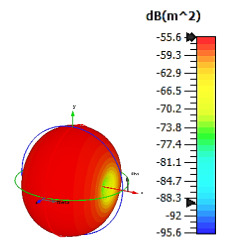
Design 3 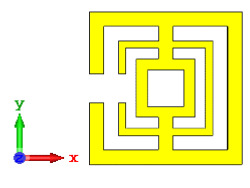	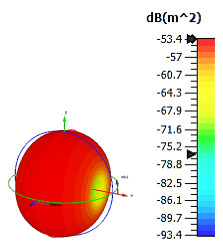	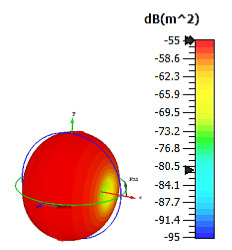	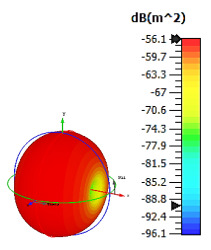

**Table 3 materials-15-04282-t003:** Bistatic scattering patterns of triangular structure with the various substrate materials.

Designs	FR-4	Rogers RT6002	Rogers RO3006
Design 1 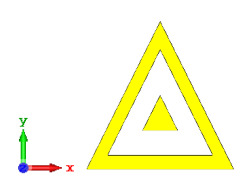	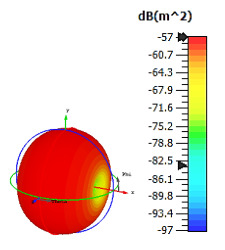	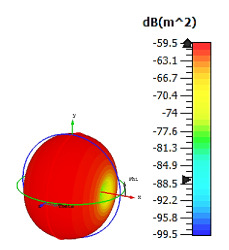	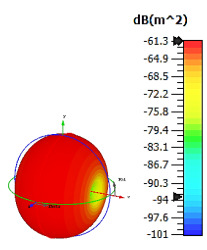
Design 2 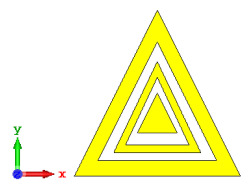	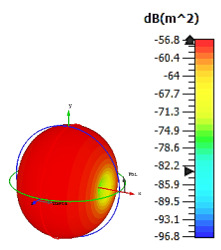	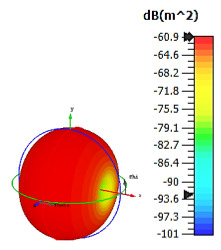	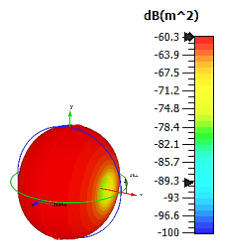
Design 3 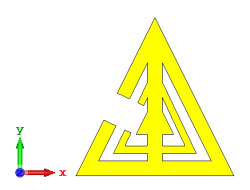	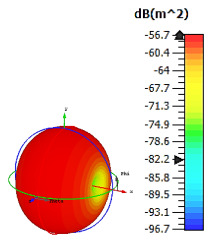	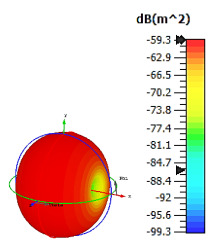	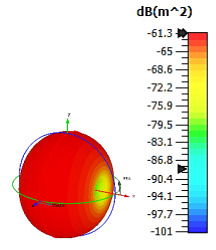

**Table 4 materials-15-04282-t004:** Bistatic scattering patterns of uniquely shaped structure with the various substrate materials.

Designs	FR-4	Rogers RT6002	Rogers RO3006
Design 1 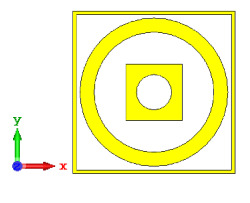	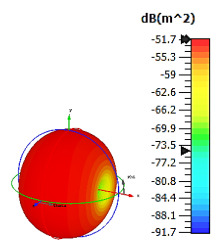	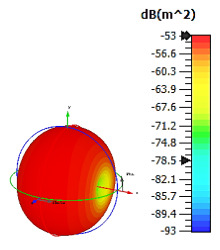	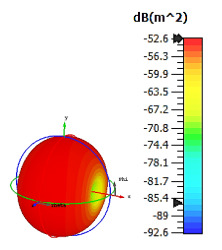
Design 2 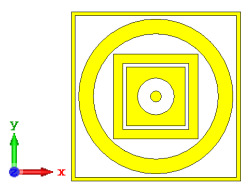	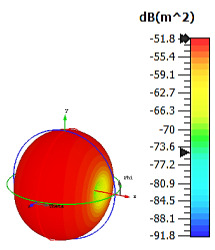	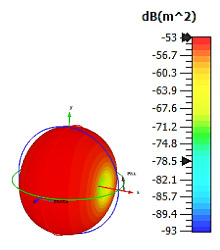	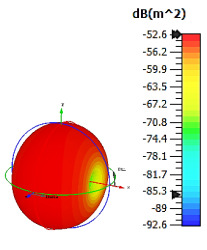
Design 3 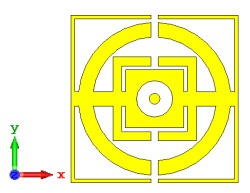	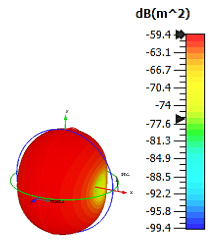	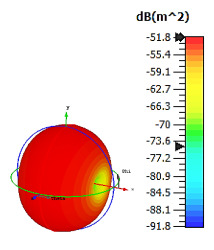	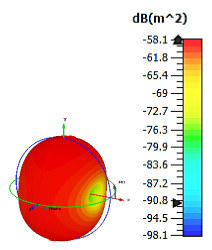

**Table 5 materials-15-04282-t005:** Dimension details of all four elements for two-bit coding metamaterials.

Descriptions	Dimension (mm)
a	5
b	5
cr1	0.4
cd1	1.0
s1	1.6
sr1	0.1

**Table 6 materials-15-04282-t006:** Coding sequences of *n* = 4 for each row.

Coding Sequence	Row 1	Row 2	Row 3	Row 4
1	1010	1010	1010	1010
2	1010	1111	1100	1001
3	1100	0011	1100	0011

**Table 7 materials-15-04282-t007:** Coding sequences of *n* = 6 for each row.

Coding Sequence	Row 1	Row 2	Row 3	Row 4	Row 5	Row 6
1	110000	001111	110000	001111	110000	001111
2	110010	101101	010000	111111	110010	010101
3	110110	101001	010000	110010	010010	011101

**Table 8 materials-15-04282-t008:** Coding sequences of *n* = 8 for each row.

Coding Sequence	1	2	3
Row 1	11011011	10011011	10010001
Row 2	10100101	11011011	00101010
Row 3	01010100	01010100	01010100
Row 4	10001010	10101001	10101001
Row 5	01001010	01101001	01011010
Row 6	00110101	00110101	10101001
Row 7	01001010	01001010	01010010
Row 8	01010101	11100111	10100101

**Table 9 materials-15-04282-t009:** Coding sequences of *n* = 12 for each row.

Coding Sequence	1	2	3
Row 1	110111011111	100111101101	111011111110
Row 2	011011111011	111001110101	110101111101
Row 3	110010110111	111110011011	101110111011
Row 4	111111111101	100111100111	011111010111
Row 5	101111010111	100110111001	101110101111
Row 6	111010111110	111101011110	110101110111
Row 7	110001101011	001010101111	101011101011
Row 8	000010111101	110111010111	011101011101
Row 9	010110111110	011001100100	101110111110
Row 10	111011110011	101011001010	110111011101
Row 11	101010101010	111011010101	101010101011
Row 12	101011011011	010101001001	011101110111

**Table 10 materials-15-04282-t010:** Comparison table of bistatic RCS reductions at 8 GHz.

Lattices	Coding Sequence	Conventional (dBm^2^)	Multi-Layer (dBm^2^)	Cuboid (dBm^2^)
4	3	−27.8	-	−17.1
6	2	−19.5	-	−12.7
8	3	−20.7	-	−7.6
12	3	−8.3	Double: −7.92 Triple: −6.39	−4.36

## Data Availability

All the data are available within the manuscript.

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
