# Peer review of "Design and Analysis of Multi-Layer and Cuboid Coding Metamaterials for Radar Cross-Section Reduction"

_materials, 2022, doi:10.3390/ma15124282_

Round 1
Reviewer 1 Report
1. As discussed in the manuscript that the thickness would also impact a lot towards the RCS values in the circular-shaped metamaterial, it is necessary for the author to give more information about the consideration why the thicknesses were fitted as 1.6 mm, 1.524 mm and 0.25 mm for these substrates.
2. It will be more convictive that providing more details about the substrate selection in the Coding metamaterial part.
3. Since the RSC of cuboid design showed that the 4 lattices case possessed the smallest RSC value, more discuss should be conducted for the necessity and the application scenario of the high order lattices case.
4. The size information of these square or circular design were not presented in the manuscript. Extrapolating from the frequency range, the fabrication process should be quite simple for these 2D SRR like design. It would be more persuasive that the experimental data was carried out.
Author Response
As attached.

Reviewer 2 Report
For a better understanding of the results of the work, it is desirable to add some data and comments:
- Are the transmitted and absorbed powers taken into account in the model, as well as the reflection from the bottom surface of the substrate?
- It is advisable to give the initial parameters of the simulation - the method of excitation of structures, the type of waves, for which zone (near, far) are diagrams built?
- It is desirable to give the resonance parameters (S12) of the used structures.
- Indicate the location of flat structures in the x,y,z axes on bistatic diagrams (Table 1).
- Bring a 2D scattering pattern in three projections to evaluate the symmetry.
- Specify monostatic direction on bistatic diagrams.
- Indicate at what frequency the bistatic diagrams are measured.
- It is advisable to show the distribution of currents in resonant structures.
- What is the reason for the choice of the geometry of resonant structures and different thicknesses of the substrates.
- Line 190. Rogers RT6002 substrate material has a 6.5 dielectric constant. Mistake?
- Why the shape of the structure does not affect to the shape of the scattering pattern?
- What correspond the resonance peaks in fig. 4?
- At what scattering angles min. RCS values?
- It is desirable to show the S12 parameters of the structures used in the monostatic mode in Fig. 5 to confirm the phase characteristics.
- Does the asymmetry of bistatic diagrams depend on the structure of the code?
- It is desirable to give S12 parameters for used lattices.
- Specify the orientation of cuboid structures in the x,y,z axes on bistatic diagrams
- What caused the asymmetry of the bistatic diagram of cuboid structures in Fig. 14?
- How will regular coding 2D structures in the lattice plane (circles, squares) affect the shape of the bistatic diagram?
- What is the physical mechanism for changing the reflected power when using different structured lattices?
Author Response
As attached.

Reviewer 3 Report
This research aims to develop coding metamaterial to reduce Radar Cross Section (RCS) values. in C- and Ku-bands applications. The authors have designed circular, square, triangular and hybrid structures to control the RCS performance. However, the manuscript is not well organized and lack of enough novelty, thus I cannot support its publication.
Some revision comments and questions:
- “Radar Cross Section Applications” in the title is recommended to be “Radar Cross Section Reduction”, because the purpose of this work is to reduce RCS, not other RCS applications. Besides, “ Multiple Layer and Cuboid Shaped Coding Metamaterial” mentioned in the title is less discussed in the text. Instead, the analysis of unit structure selection and monolayer results takes up much of the text.
- I cannot agree with the conclusion in the line 113. Since Prof. Tiejun Cui proposed coding metasurface in 2014, a large number of research work has been carried out, and many of them also study the realization of RCS reduction using coding metasurface.
- For RCS reduction, the RCS as low as possible is desirable. But in this paper, the goal seems to be the opposite. From the graph, you always pick the case with the largest RCS, but it is referred as RCS reduction in the paper. What’s wrong in this process? Is the label in the figure wrong?
- At which frequency the bistatic RCS is calculated? Besides, the law of RCS value changing with the thickness of substrate is described in this paper, but there is no relevant calculation or simulation results, which is not convincing enough.
- In the line 190, is the permittivity of Rogers RT6002 correct?
- In this paper, only simulation is carried out without experimental results which are also important to verify the results.
Author Response
As attached.

Round 2
Reviewer 1 Report
None
Author Response
Thank you for the recommendation.
Reviewer 2 Report
The approach developed in the work uses a special metasurface consisting of binary digital elements "0" or "1" of subwave length. Thus, natural resonant frequencies of individual elements of the metasurface are outside the operating frequency range. However, as it is known, a sharp change in the phase of oscillations of the resonant structure by 180 degrees occurs only when the resonance passes. As a result, Fig. 5, which shows a sharp phase change at a frequency of 10 GHz by almost 500 deg.
To clarify this discrepancy, it is still desirable to add to the text of the article the amplitude-frequency characteristic of an individual element of the metasurface (S11 or S12). The HFSS package makes this easy.
In this case, the resonant frequency characteristics of this element will determine the operating frequency range in which it is possible to effectively control the scattering of electromagnetic radiation and, in particular, the Radar Cross Section parameter.
Author Response
As attached.

Reviewer 3 Report
The authors have revised the manucript carefully according to reviewer comments. I recommend its publication.
Author Response
Thank you for the recommendation.